# Optical Tellegen metamaterial with spontaneous magnetization

Shadi Safaei Jazi [1,6], Ihar Faniayeu [2,6], Rafael Cichelero[2], Dimitrios C. Tzarouchis [3,4], Mohammad Mahdi Asgari[1], Alexandre Dmitriev [2], Shanhui Fan [5] & Viktar Asadchy [1,5] ✉

The nonreciprocal magnetoelectric effect, also known as the Tellegen effect, promises a number of groundbreaking phenomena connected to fundamental (e.g., electrodynamics of axion and relativistic matter) and applied physics (e.g., magnetless isolators). We propose a three-dimensional metamaterial with an isotropic and resonant Tellegen response in the visible frequency range. The metamaterial is formed by randomly oriented bi-material nanocylinders in a host medium. Each nanocylinder consists of a ferromagnet in a single-domain magnetic state and a high-permittivity dielectric operating near the magnetic Mie-type resonance. The proposed metamaterial requires no external magnetic bias and operates on the spontaneous magnetization of the nanocylinders. By leveraging the emerging magnetic Weyl semimetals, we further show how a giant bulk effective magnetoelectric effect can be achieved in a proposed metamaterial, exceeding that of natural materials by almost four orders of magnitude.

Most materials interact with light in the optical spectral range through electric polarization and are characterized by permittivity $\varepsilon$. The effect of magnetization in such materials, described by permeability $\mu$, at optical frequencies is negligible; however, it can be enhanced using the metamaterial paradigm[1]. In addition to the electric polarization and magnetization, materials operating in the linear regime may host bianisotropic phenomena, i.e., when magnetization can be induced by the electric component of light and polarization can be generated by the magnetic component. There are two types of bianisotropic phenomena[2,3]: reciprocal, commonly attributed to chirality, and nonreciprocal, usually referred to as the magnetoelectric, Tellegen, or magnetochiral effect. Optical chirality, occurring in various materials exhibiting reciprocity but broken parity symmetry, has been extensively studied in the literature[4]. In contrast, the nonreciprocal magnetoelectric (ME) phenomena (not to be confused with the magneto-optical effect) are extremely rare and less explored as they require materials with both broken parity symmetry and reciprocity[5,6].

The existence of the ME effect was first conceived conceptually by B. Tellegen in a composite formed by particles bearing parallel or anti-parallel static electric and magnetic dipoles[7]. Independently, in the physics community, the magnetoelectric (equivalent to Tellegen) effect was conceptualized by L. Landau[8] and predicted in $Cr_2O_3$ by I. Dzyaloshinskii[9], followed by experimental verifications[10,11]. These developments led to extensive research on ME materials and multiferroics (ferroelectric ferromagnets)[12]. As magnetism is strong at microwave frequencies, ME effects are particularly pronounced in this spectral range[13]. Low-frequency ME materials are thus considered for applications in spintronics, next-generation data storage, and ultralow power logic memory devices[12].

Since the late 1990-s, several attempts to move from natural ME materials towards artificial ME composites were made using metamaterials approach. However, meta-atoms designs were still limited to the microwave frequency range[14-17]. Furthermore, inside such three-dimensional metamaterials, the bulk isotropic Tellegen effect necessarily vanishes when uniform external magnetic bias field is used.

[1]Department of Electronics and Nanoengineering, Aalto University, P.O. Box 15500, FI-00076 Aalto, Finland. [2]Department of Physics, University of Gothenburg, Gothenburg 41296, Sweden. [3]Department of Electrical and Systems Engineering, University of Pennsylvania, Philadelphia, PA 19104, USA. [4]Meta Materials Europe, Marousi, 15123 Athens, Greece. [5]Ginzton Laboratory and Department of Electrical Engineering, Stanford University, Stanford, CA 94305, USA. [6]These authors contributed equally: Shadi Safaei Jazi, Ihar Faniayeu. ✉e-mail: viktar.asadchy@aalto.fi

Recently, two alternative ideas were conceptualized suggesting the bulk ME effect using temporally modulated permittivity and permeability at optical frequencies[18] and a multilayer metamaterial in the out-of-plane antiferromagnetic configuration[19]. However, the practical realization of both material types is highly complex, let alone in the optical regime most probably unfeasible with the current state of nanotechnology.

The observed ME phenomena at optical frequencies in known materials are negligible and thus remain elusive for realistic applications[20,21]. The exception is the topological ME effect in certain topological insulators[22–24] This effect is linked to surface states and exhibits opposite signs on either side of the surface. Consequently, its experimental investigation using light is challenging due to the canceling contributions from both interfaces of a thin film sample. Classical (i.e., non-topological) optical ME effect is vanishingly weak because both cyclotron and Larmor frequencies of electrons are typically in the GHz range[6]. Even at cryogenic temperatures already at 1 THz, the largest reported bulk ME parameter drops to $\chi = 0.02$[25]. Despite the absence of practical materials with a measurable bulk optical ME effect, the ME phenomena have been extensively theoretically studied over the last two decades. It was predicted that ME phenomena could lead to novel nonreciprocal light reflection and transmission[26], photonic topological edge states[27], PT-symmetric ME energy density[28], synthetic movement[29], directional dichroism[30,31], spin-dependent thermal radiation[32], persistent planar heat current[33], to name a few. Several phenomena, such as Fresnel drag for light[34] and photonic gauge fields[35], in principle, cannot occur in natural materials and require some structural engineering.

The possibility of an isotropic ME effect has particular importance for fundamental physics. It was pointed out in seminal works[36,37] that electrodynamic equations of isotropic ME materials have the same form as those of an axion medium. Axion is an elementary particle in the quantum field theory that was theoretically predicted to resolve two grand problems in modern physics, namely, the strong charge conjugation and parity (CP) problem[38], and the problem of the existence of the dark matter[39]. Therefore, in materials science and condensed matter physics, there has been intense interest in exploring exotic concepts of axion electrodynamics with practical ME materials[40,41]. Currently, the two suggested material classes with the ME effect for exploring axion electrodynamics are conventional crystalline[42] and topological insulators[22,43]. However, in addition to the aforementioned limitations, their ME effect is non-tunable (due to the fixed crystal lattice), making the probing of the effective axion field technologically complicated.

Here we theoretically propose a three-dimensional metamaterial with an isotropic ME effect in the visible spectral range. Its meta-atom is a nanocylinder with a ferromagnetic nanodisk in a single-domain magnetic state and a high-permittivity dielectric nanodisk supporting the magnetic Mie-type resonance. The 3D metamaterial is formed by randomly distributing (position- and orientation-wise) the nanocylinders in a host medium such as water or polymer. Due to the optimized magnetic shape anisotropy the ferromagnetic nanodisks exhibit spontaneous magnetization and the ME effect (without the need for an external magnetic bias). Random orientation of the nanocylinders in metamaterial ensures the disappearance of other types of bulk bianisotropic effects while preserving the bulk Tellegen effect, enhanced by the Lorenz–Mie-type resonances[44] in the dielectric parts of the nanocylinders[45,46]. Such a bias-free optical Tellegen metamaterial represents both qualitative (allowing tunability in space, time, and frequency) and quantitative (resonant enhancement) advancement toward realizing ME phenomena in realistic materials. We demonstrate that by using conventional materials such as cobalt and silicon, we achieve two orders of magnitude improvement of the ME effect compared to other known natural materials at room temperatures. Finally, we show that by using the emerging magnetic Weyl semimetals as the component of the meta-atoms in a metamaterial, we can enhance the ME effect almost four orders of magnitude compared to natural materials, while making bulk ME parameter the same order as the effective permittivity and permeability of the metamaterial.

## Results

### Comparison of gyrotropic, chiral, and Tellegen effects

In the approximation of linear response and weak spatial dispersion, the most general isotropic materials are characterized by the following constitutive relations that relate the displacement fields **D** and **B** to the electric and magnetic fields **E** and **H**[20]:

$$\mathbf{D} = \varepsilon_0 \varepsilon \mathbf{E} + \frac{1}{c}(\chi - i\kappa)\mathbf{H}, \quad \mathbf{B} = \mu_0 \mu \mathbf{H} + \frac{1}{c}(\chi + i\kappa)\mathbf{E}. \tag{1}$$

Here, $c$ stands for the speed of light, $\varepsilon$, $\mu$, $\kappa$, and $\chi$ are the scalar permittivity, permeability, chirality parameter, and Tellegen parameter of a continuous medium, respectively. In the case of a metamaterial, $\varepsilon$, $\mu$, $\kappa$, and $\chi$ in Eq. (1) are the effective material parameters. In what follows, we assume time-harmonic oscillations in the form $e^{+i\omega t}$. Among the four scalar material parameters, only one, namely, the Tellegen parameter, corresponds to a nonreciprocal effect, that is an effect where the Lorentz reciprocity is broken[6].

We devise the Tellegen metamaterial by designing a single meta-atom, analyzing its response inside a bulk composite of equivalent meta-atoms and finding the effective material parameters of the composite. We assume a random orientation and arrangement of the meta-atoms in the metamaterial. While this eases up the future large-scale nanofabrication, it also prompts exploiting anisotropic meta-atoms providing more design degrees of freedom. Indeed, a collection of randomly oriented anisotropic meta-atoms produces an isotropic metamaterial since any anisotropic electromagnetic effects in the individual meta-atoms will be compensated in such metamaterial[2]. For simplicity, we assume that individual meta-atoms have uniaxial (transversely isotropic) symmetry. This enables a variety of electromagnetic effects (see below) and is more accessible experimentally. Without losing the generality, we select the symmetry axis to be along the z-axis. For light incident on the meta-atom along its symmetry axis ($E_z = H_z = 0$), the induced electric **p** and magnetic **m** dipole moments can be expressed as[3]

$$\mathbf{p} = \left[\alpha_e I + \alpha_{ge} J\right] \mathbf{E} + \left[(\alpha_{\chi s} + \alpha_{\kappa s})I + (\alpha_{\kappa a} + \alpha_{\chi a})J\right] \mathbf{H},$$
$$\mathbf{m} = \left[\alpha_m I + \alpha_{gm} J\right] \mathbf{H} + \left[(\alpha_{\chi s} - \alpha_{\kappa s})I + (\alpha_{\kappa a} - \alpha_{\chi a})J\right] \mathbf{E}, \tag{2}$$

where $I$ is the two-dimensional unit matrix and $J$ is the antisymmetric two-dimensional (skew-symmetric) matrix with only nonzero components $J_{yx} = -J_{xy} = 1$. We limit our discussion to the dipolar moments since the higher-order multipoles in individual meta-atoms do not contribute to the bulk magnetoelectric effect when they are randomized in a metamaterial[47]. In Equation (2), $\alpha_e = \alpha_e^{xx} = \alpha_e^{yy}$ and $\alpha_m = \alpha_m^{xx} = \alpha_m^{yy}$ are the electric and magnetic polarizabilities describing reciprocal responses, while $\alpha_{ge}$ and $\alpha_{gm}$ are gyroelectric (due to the cyclotron orbiting of free electrons[48]) and gyromagnetic (due to the precession of electron spins[49]) polarizabilities responsible for magneto-optical effects. Here, the orientation of the external static magnetic field (or the material own magnetization) is assumed to be along the +z-direction. Such double gyrotropic response is referred to as bi-gyrotropic[50]. Polarizabilities $\alpha_{\chi s} = \alpha_\chi^{xx} = \alpha_\chi^{yy}$ and $\alpha_{\chi a}$ denote the Tellegen and so-called "synthetic moving" response of the meta-atom, whereas $\alpha_{\kappa s} = \alpha_\kappa^{xx} = \alpha_\kappa^{yy}$ and $\alpha_{\kappa a}$ describe its chirality and the so-called omega coupling, respectively[2,51].

Since the isotropic Tellegen and the other three mentioned effects, described by $\alpha_{ge}$, $\alpha_{gm}$, and $\alpha_{\kappa s}$, show up only in the cross-polarized scattered fields, we analyze separately their characteristic

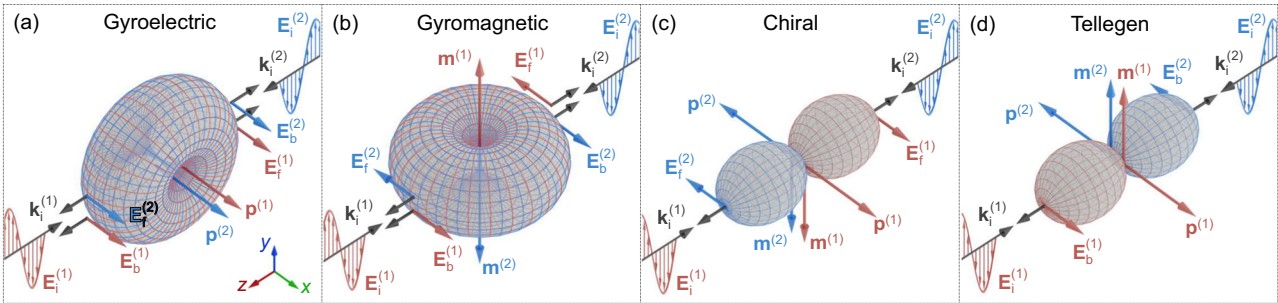

**Fig. 1 | Characteristic signatures in the cross-polarized light scattering of different electromagnetic effects in a meta-atom with a uniaxial symmetry along the $z$-axis. a** Gyroelectric magneto-optical effect. The effect can be obtained at microwaves using meta-atoms with magnetized electron plasmas and at optical frequencies using ferromagnetic meta-atoms. **b** Gyromagnetic magneto-optical effect that can be induced in the meta-atom at microwaves by exploiting ferrites. **c** Chirality effect that can be obtained in reciprocal meta-atoms with broken parity symmetry. **d** Tellegen effect that can be generated only by nonreciprocal meta-atoms with broken parity symmetry and reciprocity. Meta-atoms in **a**, **b**, and **d** have static magnetizations along the $+z$-direction.

signatures. Using Eq. (2), we qualitatively illustrate in Fig. 1 how the four selected effects influence light scattering by a uniaxial meta-atom. Light scattering with the same (co-) polarization as that of the incident light is not shown for clarity. In each subfigure, we depict dipole moments induced by the incident fields $\mathbf{E}_i$ and $\mathbf{H}_i$ and their radiation patterns for two scenarios: light incident along the $-z$-direction marked with a superscript '(1)' (red arrows) and the $+z$-direction marked with a superscript '(2)' (blue arrows). Subscripts 'f' and 'b' refer to the forward and backward scattering, respectively. The black arrows denote the wave vectors $\mathbf{k}$ of the incident and scattered waves. As is seen in Fig. 1a, gyroelectric response of a meta-atom results in a cross-polarized (with respect to the incident electric field) electric dipole that radiates symmetrically cross-polarized light in the backward (magneto-optical Kerr effect) and forward (magneto-optical Faraday effect) directions. Notably, the light scattering is independent of the incidence direction. According to Fig. 1b, a gyromagnetic meta-atom, likewise, scatters cross-polarized light with the same intensity into backward and forward directions, but the phase of light now depends on the illumination.

Chirality in a meta-atom is responsible for reciprocal cross-polarization conversion in transmission (Fig. 1c). Here, the radiation patterns for the two illuminations are oppositely oriented but both correspond to a Huygens'-type pattern with no backward and maximum forward radiation[52]. The sense of polarization rotation (clockwise or anti-clockwise with respect to the propagation direction) is independent of the illumination direction confirming reciprocity of a chiral meta-atom[3].

Finally, a uniaxial meta-atom with the Tellegen response behaves somewhat oppositely to a chiral meta-atom: regardless of the illumination direction, the induced dipole moments do not scatter light in the forward direction (Fig. 1d). Instead, they scatter asymmetrically backwards. Note that this does not violate the optical theorem[53] since the zero forward scattering occurs for the cross-polarized portion of scattered light. Thus, the nonreciprocity in a Tellegen meta-atom manifests itself in the cross-polarized reflection but does not appear for transmitted light. Compared to gyroelectric and gyromagnetic effects where both Kerr and Faraday rotations occur, the Tellegen effect hosts only the former. Due to this feature, even bulk Tellegen material slabs do not produce cross-polarized light in transmission and can be probed only in reflection[20].

By comparing Eqs. (1) and (2), we observe that in a bulk isotropic metamaterial with randomly oriented meta-atoms with statistical averaging of their scattered fields only the effects characterized by $\alpha_e$, $\alpha_m$, $\alpha_{\chi s}$, and $\alpha_{KS}$ survive[2]. Therefore, to design a purely Tellegen metamaterial, we must ensure that all the meta-atoms have zero $\alpha_{KS}$ polarizability.

## Design of a Tellegen meta-atom

The challenge to reach an isotropic Tellegen metamaterial is to produce a space-inhomogeneous external magnetization (i.e., each meta-atom should be magnetized externally) with a sufficiently strong field amplitude. This is a bottleneck of all the known microwave designs of Tellegen meta-atoms[14–17]. If the external magnetization is uniform in space, the bulk Tellegen effect would be fully compensated because magnetization in each meta-atom will be random with respect to its geometry. In other words, isotropy of the metamaterial forbids uniform external bias fields. Nevertheless, local magnetization of microwave meta-atoms is, in principle, possible if they are equipped with non-conducting permanent magnets, such as barium ferrite $BaFe_{12}O_{19}$[54,55], but such implementations have not yet been reported. Another challenge is that at optical frequencies the magneto-optical effects exploited for the generation of the effective Tellegen response are inherently very weak[6] since their resonances occur in the microwave region for a typical magnetic flux density in the order of a few Tesla.

In order to address both challenges, we propose a meta-atom with the geometry shown in Fig. 2a.

The proposed meta-atom is a nanocylinder with one part ferromagnet or ferrimagnet (ferrite) and one part high-permittivity semiconductor or dielectric. As an example, we select cobalt as a ferromagnet having one of the largest magneto-optical parameters $Q = |\varepsilon_{xy}/\varepsilon_{xx}| \sim 0.02$ in the visible frequency range[56]. The ferromagnet must be in a single-domain state[57], which puts a constraint on its size and aspect ratio $\delta = h_{Co}/D$. Although nanoscale ferromagnets with high aspect ratios are superior for attaining the single-domain configuration, they are usually challenging to fabricate. We thus keep $\delta = 1$. There is an upper limit on the diameter of the ferromagnetic nanocylinder in a single-domain state since the magnetostatic energy scales up faster with its size (proportional to its volume) than the energy required to create a domain wall (proportional to the nanocylinder cross-section)[58]. For a chosen cobalt nanocylinder the maximum diameter is $D_{max} = 96$ nm (see Supplementary Section 1 and Supplementary Fig. 2). There is also a lower size limit as the thermal energy $k_B T$ might become sufficiently large to flip the magnetization produced by the magneto-crystalline and shape anisotropies in the smaller nanocylinder[59] (here $k_B$ is the Boltzmann constant and $T$ is the temperature of the nanocylinder). We estimate the minimum diameter of a cobalt cylinder to be $D_{min} = 7.4$ nm at the room temperature of $T = 300$ K (see Supplementary Section 2 and Supplementary Fig. 3). Importantly, these theoretical limits are rather close to the experiment[60]. Here we choose the cobalt nanocylinder having $D = D_{max}$) to reach the optical resonance in the visible spectrum for the whole meta-atom.

The ferromagnetic nanocylinder in a single-domain state possesses permanent static magnetization close to the saturation

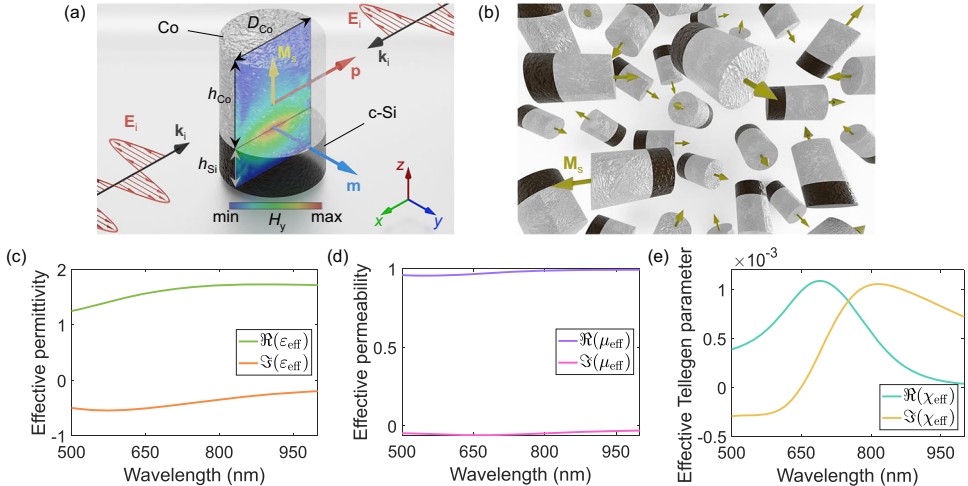

**Fig. 2 | Cobalt-based optical Tellegen meta-atom and isotropic Tellegen metamaterial with its effective material parameters. a** Uniaxial Tellegen meta-atom operating in the visible. The upper nanocylinder is a ferromagnet (Co) in the single-domain state. Saturation magnetization direction $\mathbf{M_s}$ defines the sign of the Tellegen polarizability. The lower nanocylinder is a high-index dielectric (Si) operating at the magnetic Mie-type resonance ($h_{Co} = 96$ nm, $h_{Si} = 43$ nm, $D_{Co} = 96$ nm). Incident $y$-polarized electric field $\mathbf{E_i}$ in the standing-wave configuration

induces, among others, a magnetic dipole moment $\mathbf{m}$ along the $y$-direction (for clarity, not all induced moments are shown). The field map shows the magnetic field distribution inside the meta-atom at resonance ($\lambda = 750$ nm). **b** Schematics of the bulk optical Tellegen metamaterial with cobalt-silicon meta-atoms. The yellow arrows depict the orientations of the local magnetization in single-domain ferromagnetic nanocylinders. Effective parameters of the metamaterial in **b**: (**c**) relative permittivity, (**d**) relative permeability, and (**e**) bulk Tellegen parameter.

magnetization of cobalt $M_s = 1.44 \times 10^6$ A/m. The magnetization direction (up or down) determines the sign of the effective Tellegen polarizability $\alpha_{\chi s}$. It can be defined upon the fabrication. Fixed magnetization along the nanocylinder axis can be obtained through the magnetocrystalline, shape, and/or the perpendicular magnetic anisotropies of the nanocylinder[57,61]. While the shape anisotropy is generated when the nanocylinder has a high aspect ratio ($\delta > 1$), the magnetocrystalline anisotropy is an intrinsic property due to the cobalt lattice. We assume the magnetocrystalline anisotropy to be parallel to the nanocylinder's axis, thus, the coercive field can reach up to $H_c = 6.8$ kOe (see Supplementary Section 3 and Supplementary Fig. 5). Such a coercive field ensures that the magnetization in each meta-atom is securely fixed to its geometry, which in turn means that a randomly oriented mixture of the meta-atoms will have an uncompensated bulk Tellegen response. Addressing the generally weak magneto-optical response at optical frequencies, we note that in the single-domain state, the magnetization is the highest $M \approx M_s$, leading to the highest magneto-optical effect for a given ferromagnet[6]. Additionally, the bottom nanocylinder made of a high-permittivity semiconductor (crystalline Si, Fig. 2a) allows us to substantially enhance the Tellegen polarizability $\alpha_{\chi s}$ of the meta-atom since it operates at the magnetic Mie-type resonance[52].

Our method of achieving local magnetization through the anisotropies of nanocylinders is drastically different from the approach taken in the recently proposed uniaxial multilayer Tellegen metamaterial[19]. Achieving local magnetization in such a multilayer film, which features an antiferromagnetic out-of-plane configuration, presents a considerably greater challenge because maintaining opposite magnetization vectors in neighboring domains is energetically unfavorable.

To provide a qualitative picture of the magnetoelectric effect in the meta-atom, we excite it by a standing wave with the electric field polarized along the $y$-axis and with the node of the magnetic field located at the center of the meta-atom (Fig. 2a). Note that such illumination has a different wavevector direction $\mathbf{k_i}$ compared to the one in Fig. 1d. We choose it here for simplifying the analysis as it minimizes the effects of spatial dispersion in the meta-atom. While in the dipolar approximation, there are several moments excited in the meta-atom by the electric field, in our discussion and in the figure, for clarity, we

focus only on the moments that lead to the ME effect. Due to the antisymmetric (gyroelectric) part of the nanocylinder permittivity tensor, the incident electric field induces an oscillating dipole moment $\mathbf{p}$ orthogonal to both incident field and magnetization $\mathbf{M_s}$[6]. This dipole moment generates the curl of the high-frequency magnetic field inside the semiconductor part of the nanocylinder according to Ampere's law. This magnetic field, in turn, induces a non-zero optical magnetic dipole moment $\mathbf{m}$ inside the semiconductor nanocylinder. The chosen size of the semiconductor nanocylinder supporting magnetic Mie-type optical mode ensures this induced magnetic dipole moment is brought to the resonance. To support this qualitative description, we plot the magnetic field distribution (complex amplitude) at the cross-section of meta-atom at resonance ($\lambda = 750$ nm) (Fig. 2a) obtained via full-wave simulations. A magnetic-dipole-like field distribution indeed emerges at the interface between the two nanocylinders. The Mie-type resonant mode in Si is hybridized due to the close proximity of the cobalt nanocylinder. Thus, the incident $y$-polarized optical electric field induces the $y$-directed optical magnetic moment in the meta-atom, which is the essence of the optical ME (Tellegen) effect. Conversely, an incident optical magnetic field would generate an electric dipole moment. The semiconductor nanocylinder of the meta-atom has the additional role of breaking the parity symmetry that a pure cobalt nanocylinder would have. The magnetic point group of the meta-atom $C_{\infty v}(C_\infty)$[62] includes only the symmetry operations $\underline{m}_x$, $\underline{m}_y$, and $4_z$. Here, $\underline{m}_x$ denotes a mirror symmetry with respect to the $x$ axis followed by a time reversal and $4_z$ denotes a four-fold rotational symmetry[63]. {Thus, both parity (space-inversion) symmetry and reciprocity are broken in the meta-atom, as required for a Tellegen metamaterial constituent. Due to its magnetic point group, the meta-atom has only 9 non-zero independent polarizability components: $\alpha_e$, $\alpha_e^{zz}$, $\alpha_m$, $\alpha_m^{zz}$, $\alpha_{ge}$, $\alpha_{gm}$, $\alpha_{\chi s}$, $\alpha_\chi^{zz}$, and $\alpha_{\kappa a}$ (see Supplementary Section 4 and Supplementary Figure 6). The meta-atom cannot possess chirality due to the symmetry, that is $\alpha_{\kappa s} = 0$. Moreover, as discussed above, the gyrotropy effects will disappear upon statistical averaging of the scattered fields in a metamaterial with randomly oriented meta-atoms.

## Bulk optical Tellegen metamaterial and giant Tellegen response
By using the full-wave simulations, we calculate the traces of the electric, magnetic, and magnetoelectric polarizability tensors. Next,

we use the Maxwell Garnett mixing rule to determine the macroscopic effective-medium parameters of a bulk Tellegen metamaterial (Fig. 2b) consisting of a random arrangement of the designed meta-atoms (see "Methods" section, Supplementary Section 5, and Supplementary Fig. 7). For simplicity, we assume the host medium to be air (a more practical scenario with a dielectric host medium would have qualitatively similar results with the only change of increased effective permittivity). We choose the volumetric concentration of the meta-atoms $N = 1.45 \times 10^{20} \, \text{m}^{-3}$, which is equivalent to the volume fraction $N_V = 0.15$ (ratio between the volume occupied by the meta-atoms and the total volume of the metamaterial). Figure 2c, d plot the calculated effective permittivity $\varepsilon_{\text{eff}}$, permeability $\mu_{\text{eff}}$, and Tellegen parameter $\chi_{\text{eff}}$ of the Tellegen metamaterial. The relative effective permittivity has the real part $\Re(\varepsilon_{\text{eff}}) > 1$ in the visible wavelength range due to the dominant electric polarizability $\alpha_e$ (see Supplementary Fig. 7) and relatively high density of the meta-atoms. In contrast, $\mu_{\text{eff}} \approx 1$ due to the small magnetic polarizability of the meta-atoms. Although it is resonant, the peak appears flattened when compared to the relative permeability of air. The nonzero imaginary parts of $\varepsilon_{\text{eff}}$ and $\mu_{\text{eff}}$ are due to the absorption loss in cobalt and silicon nanodisks as well as the radiation loss due to the random mixing of the meta-atoms in the metamaterial (the dominant mechanism is the absorption loss in cobalt). Likewise, the effective Tellegen parameter is resonant and reaches $\chi_{\text{eff}} \sim 10^{-3}$. Such a high value is two orders of magnitude greater than the observed bulk ME effect in natural materials at room temperatures[64,65]. We stress that the proposed ME metamaterial has important fundamental advantages over natural materials. By adjusting its meta-atom geometry and constituent materials we are potentially able to tune its resonant wavelength within the optical range. Moreover, the possibility of structuring the metamaterial (making it nonuniform in space and/or time) has important implications for generating and analyzing the effective nonuniform axion field[40,41] and related effects[37,43,66–68]. Nonuniformity in the axion field is essential since this field enters into the Maxwell equations only through derivatives with respect to time or space[69].

Although the cobalt-based Tellegen metamaterial (Fig. 2b) readily provides a record-high bulk ME response, we further extend our design approach to other magneto-optical materials. Specifically, the anomalously strong magneto-optical response was recently predicted and experimentally confirmed in the so-called magnetic Weyl semimetals[70,71]. Weyl semimetals exhibit conduction and valence bands that touch at discrete points (Weyl nodes) in momentum space, i.e., the Brillouin zone. Each Weyl node acts as a source or sink of Berry curvature, a quantity in quantum mechanics that describes the phase evolution of quantum states. Weyl nodes come in pairs of opposite chirality (analogous to the concept of 'handedness') and are topologically protected[72], meaning that they cannot be annihilated without a significant change in the system's properties. In the case of magnetic Weyl semimetals, the Weyl nodes of opposite chiralities are at the same energy but separated from one another in the momentum space. Weyl semimetals have been recently put forward as a means to generate compact optical isolators, orbital angular momentum detectors, and nonreciprocal thermal emitters among many others[69].

We consider the topological semimetal EuCd$_2$As$_2$ experimentally characterized in[73]. For the calculation of its antisymmetric dielectric tensor, we use the same material parameters as in ref. 74. The frequency dispersion of the diagonal and off-diagonal permittivity components of the Weyl semimetal is plotted in Supplementary Fig. 8 at room temperature (see also Supplementary Section 6). For simplicity, in our qualitative analysis, we do not consider the Fermi arcs existing in Weyl semimetals (unique surface states that connect the projections of different Weyl points onto the surface Brillouin zone). The giant magneto-optical response of the Weyl semimetal occurs at the mid- and far-infrared frequency ranges. The geometry of the designed Weyl-based Tellegen meta-atom is shown in Fig. 3a. Similarly to the cobalt-based meta-atom, it is built out of the upper magneto-optic (Weyl semimetal) and lower high-permittivity (amorphous silicon[75]) nano-cylinders. The diameter $D_{\text{Weyl}} = 530 \, \text{nm}$ of the Weyl nanocylinder with unit aspect ratio is chosen under the upper limit (around 1 μm) for the single-domain magnetization in similar semimetals[76]. Figure 3a also features the optically resonant magnetic field distribution (complex amplitude) when the meta-atom is illuminated by an incident standing wave. Similarly to the case in Fig. 2a, one observes the magnetic-dipole-like field distribution. Next, we determine the six polarizability components of the Weyl meta-atom (see "Methods" section, Supplementary Section 7, and Supplementary Fig. 9) and, using the Maxwell Garnett mixing rule, find the effective material parameters of Weyl-based mid-infrared Tellegen metamaterial. The meta-atoms are randomly oriented and distributed within the host medium (air) (Fig. 3b).

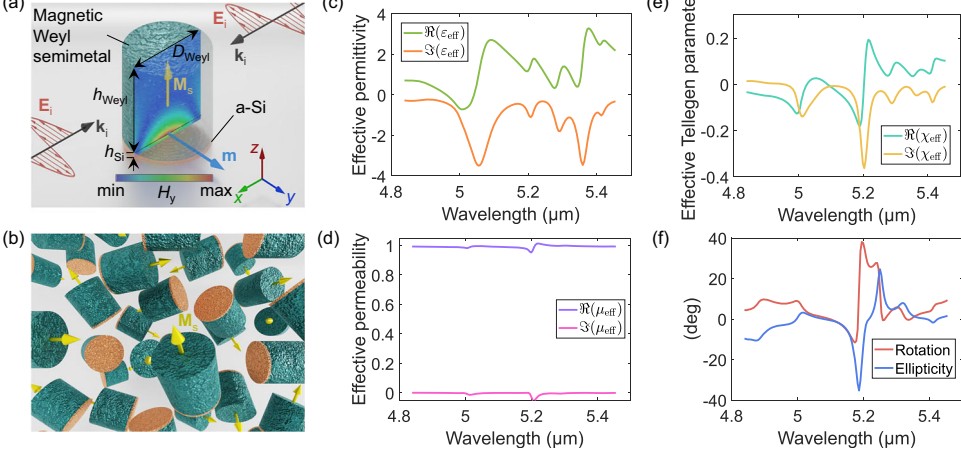

**Fig. 3 | Weyl-semimetal-based mid-infrared Tellegen meta-atom and isotropic metamaterial based on it with the effective material parameters. a** Tellegen meta-atom for mid-infrared wavelengths. The upper nanocylinder consists of a ferromagnetic (Weyl semimetal EuCd$_2$As$_2$) in the single-domain magnetization state. The lower nanocylinder is amorphous silicon. The dimensions of the meta-atom are $h_{\text{Weyl}} = 530 \, \text{nm}$, $h_{\text{Si}} = 26 \, \text{nm}$, and $D_{\text{Weyl}} = 530 \, \text{nm}$. Incident y-polarized electric field induces a magnetic dipole moment along the y-direction. The field map plots the magnetic field distribution inside the meta-atom at resonance ($\lambda = 5 \, \mu\text{m}$). **b** Schematics of the bulk mid-infrared Tellegen metamaterial with Weyl-silicon meta-atoms. The yellow arrows depict the orientations of the local magnetization in single-domain ferromagnetic nanocylinders. **c–e** Effective parameters of the metamaterial in **b**. **f** Kerr rotation and Kerr ellipticity calculated for plane waves incident on the planar interface between air and metamaterial in **b**.

We choose the volumetric concentration of the meta-atoms $N = 1 \times 10^{18} \mathrm{m}^{-3}$, equivalent to the volume fraction of $N_V = 0.12$. The effective material parameters of the metamaterial are depicted in Fig. 3c–e. We note the presence of multiple Lorentzian resonances in all the parameters as the considered frequency range is below the plasma frequency of the Weyl semimetal. Due to the anomalous magneto-optical response of the semimetal, the Tellegen bulk parameter $\chi_{\mathrm{eff}}$ at 5.2 μm reaches the value of 0.36, four orders of magnitude higher than any known material. Remarkably, at the same wavelength, the effective permittivity and permeability are weakly resonant at $\varepsilon_{\mathrm{eff}} = 1.28 - i1.04$ and $\mu_{\mathrm{eff}} = 0.96 - i0.05$, comparable in amplitude to the Tellegen parameter. This giant ME parameter potentially elevates the utility of Tellegen media, ushering it into practical applications. To demonstrate one of the implications of the giant Tellegen response, we consider the magneto-optical Kerr effect for plane waves incident on the planar interface between air and designed Weyl-based metamaterial[6]. As discussed above, the Tellegen effect leads to the Kerr rotation and ellipticity that are independent of the illumination side of the isotropic metamaterial. Figure 3f depicts both quantities in the studied wavelength range. The Kerr rotation and ellipticity follow the Kramers-Kronig relation and exhibit resonance at the wavelength of 5.2 μm, reaching the record values in the considered optical range (for typical magneto-optical materials, the Kerr rotation does not exceed 1 deg in the optical frequency range[50]).

### Interplay between Tellegen and gyrotropic responses

Next, we analyze the contribution of the Tellegen and gyrotropic effects in the optical Tellegen metamaterial and visualize how different structural blocks in the designed meta-atom affect the Tellegen response. We do this on the example of Weyl-based Tellegen meta-atoms (Fig. 3a). As mentioned above, the gyrotropic effect present in each meta-atom cannot contribute to the macroscopic gyrotropy of the metamaterial due to meta-atoms' random orientations. Next, we confirm this statement. Since the full-wave simulations of a three-dimensional metamaterial are very challenging (the simulation domain is large, and one needs a large set of such simulations for obtaining meaningful results using statistical averaging), we instead analyze a single layer of meta-atoms (metasurface). We further simplify the problem by assuming a periodic arrangement of the meta-atoms along the x- and y-directions with the period of 2 μm ("Methods" section). Each supercell includes four meta-atoms (Fig. 4a). By changing the mutual orientation of the four meta-atoms within the supercell, we emulate both oriented and random arrangements of the same meta-atoms as in a bulk metamaterial. Following the results shown in Fig. 1, we can define the two auxiliary parameters $R_{\mathrm{Xs}}$ and $R_{\mathrm{ge}}$ that allows us to separate the

Tellegen and gyroelectric (same applies to gyromagnetic) effects from one another. Although both effects lead to the cross-polarized reflection, the phases of such reflections for two opposite illuminations (1) and (2) are equal in the case of gyrotropic and opposite in the case of the Tellegen effect. Therefore, we define the gyroelectric reflection parameter as $R_{\mathrm{ge}} = R_{xy}^{(1)} + R_{xy}^{(2)}$, while the Tellegen reflection parameter is given by $R_{\mathrm{Xs}} = R_{xy}^{(1)} - R_{xy}^{(2)}$. Here, $R_{xy}^{(1)}$ and $R_{xy}^{(2)}$ stand for complex-valued cross-polarized reflection coefficients from the array of meta-atoms (the reference plane is located at the geometric center of the array, i.e. at $z = 0$) when the incident wave propagates along the $-z$- and $+z$-directions, respectively.

First, the four meta-atoms are all oriented along the z axis (Fig. 4a). This is equivalent to an oriented (anisotropic) three-dimensional Tellegen metamaterial. We then calculate the reflection parameters $R_{\mathrm{ge}}$ and $R_{\mathrm{Xs}}$ for the forward and backward incidences (Fig. 4a). In this case, in addition to the Tellegen, the array exhibits an uncompensated and dominant gyroelectric response. However, by making the metamaterial become isotropic through randomizing its meta-atoms (having an equal number of meta-atoms looking up and down inside the supercell), the gyroelectric effect $R_{\mathrm{ge}}$ vanishes, while the Tellegen response $R_{\mathrm{Xs}}$ is nearly unchanged due to the local magnetization in the meta-atoms (Fig. 4b). Thus, such an isotropic configuration represents a purely Tellegen metamaterial.

Next, we check how the metamaterial behaves if the magnetization in each meta-atom is independent of its orientation, i.e., equal for all Tellegen meta-atoms. This occurs when the meta-atom is not in the single-domain magnetic state and the magnetization can be viewed as global (Fig. 4c). Although each meta-atom has the Tellegen response, it vanishes for the metamaterial as a whole ($R_{\mathrm{Xs}} = 0$). This explains why despite that the design of individual Tellegen meta-atoms is known for a long time[14–17], bulk Tellegen metamaterials based on them could not be proposed. However, the single-domain magnetic state in a meta-atom alone is not enough for the bulk Tellegen response. Additionally, we need to break the inversion (parity) symmetry of the meta-atom. Here, we have a Si nanocylinder on one side of the meta-atom that does this. Once we remove the silicon nanocylinders, the Tellegen response disappears both in oriented (Fig. 4d) and random arrangements (Fig. 4e).

### Discussion

We proposed a three-dimensional (bulk) Tellegen (ME) metamaterial with an isotropic ME effect in the optical regime. The individual meta-atoms exhibit spontaneous magnetization and, therefore, do not require an external bias field. We further demonstrate that using an emerging class of magnetic Weyl semimetals, the effective ME response can reach values on par with those of permittivity and

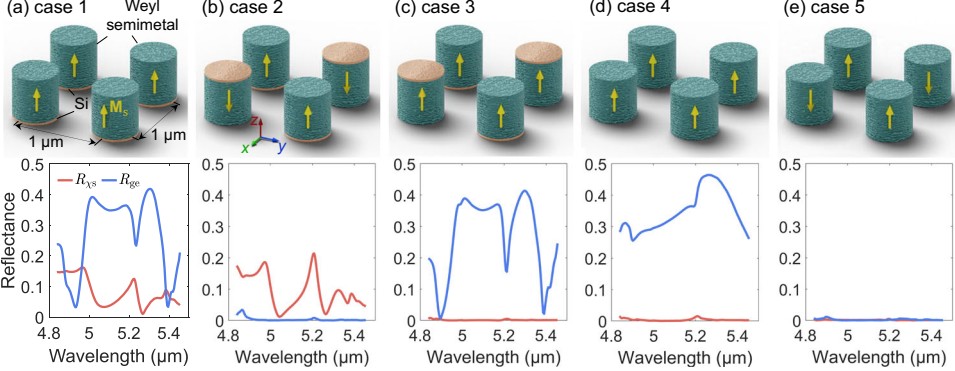

**Fig. 4 | Periodic two-dimensional arrays of meta-atoms equivalent in their optical reflection properties to bulk metamaterials.** In each illustration, only one supercell is shown. **a** Oriented and (**b**) effectively random arrays of Tellegen meta-atoms, equivalent to anisotropic and isotropic Tellegen metamaterials, respectively. The macroscopic Tellegen response vanishes if the magnetization in the meta-atoms is global which is the case in **c** or if the meta-atoms have inversion symmetry which is the case in **d** or **e**.

permeability. Realized experimentally, such a giant ME response would enable developments and applications in the field that so far has remained purely theoretical[26–29,31–33]. The proposed concept can be further extended to create anisotropic Tellegen metamaterials, in particular, to the "synthetically moving" media[3,30,34,35], and enable their implementation in the optical regime. An important fundamental implication of the proposed Tellegen metamaterial is in the electrodynamics of hypothetical axion matter. The axion electrodynamics is characterized by an additional term $\mathcal{L}_\theta = \theta(r,t)e^2\mathbf{E}\cdot\mathbf{B}/(2\pi h)$ appearing in the Maxwell Lagrangian[69] ($\theta$ is the axion field, $e$ is the electron charge, and $h$ is the Planck constant). Applying the duality transformations to these equations, one can obtain the conventional Maxwell equations, but the constitutive relations get the new form, precisely as in Eq. (1)[69,77]. This circumstance provides us with the opportunity to explore light propagation in arbitrary axion fields $\theta(r, t)$ by exploring light propagation in the Tellegen metamaterial with the Tellegen parameter $\chi(r, t)$ that can be straightforwardly tuned experimentally by design. This is in sharp contrast with recently proposed condensed-matter platforms emulating axion electrodynamics. These materials can model only a constant axion field due to their fixed crystalline structures (topological and conventional insulators[22,42,43]) or the axion field with a specific linear dependence on time and coordinate (bulk Weyl semimetals[69]). This obviously limits their ability to fully reproduce the behavior of a true axion field. Therefore, the proposed class of designer Tellegen metamaterials can offer novel avenues towards the first experimental implementations of various concepts related to axion matter, including dyon quasiparticles[66], anyon statistics[66], the Witten effect[37], and axionic polaritons[43,67]. For example, by positioning an electric monopole charge near a bulk Tellegen metamaterial, one can excite a peculiar charge distribution inside the metamaterial that would correspond to a combination of effective electric and magnetic monopole charges (dyon charge)[78].

## Methods

### Polarizabilities calculation
We calculate the polarizabilities of the Tellegen meta-atoms depicted in Figs. 2a and 3a using the full-wave simulations. The frequency dispersions for cobalt (diagonal and off-diagonal components) and crystalline silicon are taken from the experimental data in refs. 56 and 79, respectively. We employ the rigorous approach based on the calculation of the high-frequency polarizations and magnetizations induced in the meta-atom under given excitation[80]. To independently determine different polarizability components, the meta-atom is illuminated by a set of specific standing waves[81]. The calculated polarizabilities can be found in Supplementary Figs. 7 and 9. By calculating the traces of the three polarizability tensors $2\alpha_e + \alpha_e^{zz}$, $2\alpha_m + \alpha_m^{zz}$, and $2\alpha_{\chi s} + \alpha_\chi^{zz}$ and using the Maxwell Garnett mixing rule (see Section 7.2.2 in Ref. 2), we determine the macroscopic effective-medium parameters of a bulk Tellegen metamaterial consisting of the designed meta-atoms.

### Simulation
The responses of the two-dimensional arrays of Tellegen meta-atoms in Fig. 4 were simulated using commercial COMSOL Multiphysics software based on the Frequency Domain Source Sweep (FDSS) solver. The periodic boundary conditions were used to create the effect of an infinite simulation domain for the structure. The polarization of incident light was defined in the incident port, and the reflection and transmission spectra were calculated using the scattering parameters of COMSOL.

## Data availability
The authors declare that the data supporting the findings of this study are available within the paper, in the Supplementary file, and are available from the corresponding authors upon request.

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

## Acknowledgements

The authors would like to thank Prof. Sergei Tretyakov and Dr. Cheng Guo for the fruitful discussions. S.S.J., M.M.A., and V.A. acknowledge the Academy of Finland (Project No. 356797). V.A. acknowledges the PREIN Flagship (Project No. 320167). S.F. received support from the MURI project from the U.S. Air Force of Office of Scientific Research (Grant No FA9550-21-1-0244). I.F., R.C., and A.D. acknowledge the Swedish Research Council for Sustainable Development (Formas) (Project No. 2021-01390).

## Author contributions

V.A. conceived the idea and performed theoretical calculations. V.A., I.F., D.C.T., and R.C. determined the optimal topology of the meta-atom. S.S.J., I.F., V.A., and M.M.A. performed full-wave simulations. A.D., S.F., and V.A. supervised the work. All the authors contributed to the discussions of the results and the manuscript preparation.

## Competing interests

The authors declare no competing interests.
