## [Peer Review File · Nature Communications]

REVIEWER COMMENTS

Reviewer #1 (Remarks to the Author):

The authors theoretically propose a metamaterial showing a remarkably large isotropic Tellegen effect. They modeled an array of randomly oriented meta-atoms consisting of a conjunction of a semiconductor with a ferromagnet. They show that the Tellegen effect in this metamaterial is larger by two orders at visible frequencies and four orders at far-infrared than in natural materials. This large effect is very impressive and potentially useful for applications in spintronics. The possibility of locally tuning the Tellegen coefficient is. Moreover, the authors' construction is creative and simpler than a naïve construction based on Tellegen meta-atoms, which itself has zero magnetization and zero electric dipole moment. The authors used the fact that the magnetization and electric dipole moment of a single meta-atom does not need to be zero in a randomly oriented array, because those moments cancel out on average, while the Tellegen coefficient averages to a finite value. The manuscript is clearly written and easy to read. I recommend this paper for publication in Nature Communications.

I have one minor comment for the authors – it is unclear why the meta-atom has a parity-time (PT) symmetry. Does the parity mean mirror parity or spatial-inversion parity?

Reviewer #2 (Remarks to the Author):

I studied the article "Optical Tellegen metamaterial with spontaneous magnetization" by Jazi and colleagues.

The authors describe an interesting proposal to realize an isotropic Tellegen response based on a particle with two components: a piece with permanent magnetization (responsible for the nonreciprocal response) and a resonant Si cylinder supporting a magnetic Mie resonance. They show that by symmetry such a nanoparticle can have a Tellegen uniaxial response (as well as other types of dipolar couplings, such as a gyrotropic-type response). They argue that after the random mixing of the nanoparticles, only the Tellegen and chiral isotropic pieces can survive due to symmetry (the chiral term vanishes by symmetry). They propose two designs that yield a sizable Tellegen response at optical frequencies. Furthermore, they nicely highlight the role of the parity and time-reversal symmetries, showing that only designs that break simultaneously the two symmetries, but that respect the PT symmetry can give the desired material response.

On the overall, this is nicely written and timely article and I support its publication in Nat. Communications, essentially in its present form. The article is important as, to my knowledge, it is the first fully-fledged proposal of a Tellegen-type metamaterial at optical frequencies. Tellegen responses are interesting conceptually and for nanophotonic applications.

I have a few minor comments:

page 3: it is strange to designate α_{ka} as the anti-symmetric Tellegen piece as by definition a Tellegen response must be a symmetric tensor. Does α_{ka} describes a moving-medium coupling?

page 5, Fig. 2a): the figure shows a plot of H_x . Is this really the relevant quantity to show? (the induced magnetic dipole is along-y). Perhaps it would be better to show the electric currents in the considered plane?

page 5: in the design of Fig. 2 what controls the material loss? The intrinsic loss of Co? Or the loss of Si? Radiation loss due to the random mixing? Please comment on the article.

page 8: the article ends by arguing that the considered developments may lead to experimental implementations of concepts related "axion matter, dyons, Witten effect, etc". I recommend that if there is indeed some interesting optics analogy/implication that it is elaborated a bit more for one of the particular items. I do not find specially useful/appealing to present a long list of items that appear to be completely detached from any specific physical setup/electromagnetic effect that can be understood by a reader.

Reviewer #3 (Remarks to the Author):

The manuscript authored by S.S. Jazi et al. describes a possible physical implementation of an isotropic Tellegen medium operating at optical frequencies. This is achieved through a meta-atom design comprising a spontaneously magnetized ferromagnetic nanocylinder attached to a dielectric nanocylinder.

The primary idea behind this research is to use the gyroelectric response displayed by the ferromagnetic nanocylinder to excite a Mie-resonant magnetic mode within the dielectric nanocylinder. Although the gyrotropic electric dipole response is

perpendicular to the high-frequency electric field, the magnetic response of the dielectric nanocylinder induced by the gyroelectric dipole aligns with the electric field. Because of that and due to the non-reciprocity of the gyroelectric response, the meta-atom exhibits characteristics similar to those of an anisotropic Tellegen particle. A metamaterial composed of such randomly oriented meta-atoms forms an isotropic Tellegen medium.

In the section discussing existing proposals for realizing Tellegen media targeting microwave frequencies, the authors assert that the main challenge in such realizations lies in creating a local magnetic biasing field. However, I disagree with this viewpoint, as the mentioned difficulty could be effectively addressed by equipping each meta-atom with a small permanent magnet attached to it. Consequently, I suggest that the manuscript authors could improve this discussion by outlining other -- and possibly more important -- differences between their proposed implementation and the existing approaches.

We have revised the manuscript in accordance to the reviewers helpful comments and suggestions. Our responses to reviewers' comments and description of changes in the manuscript are appended below, following the original comments, and for easier reading all new and changed parts are highlighted in red in the revised manuscript.

Referee 1

The authors theoretically propose a metamaterial showing a remarkably large isotropic Tellegen effect. They modeled an array of randomly oriented meta-atoms consisting of a conjunction of a semiconductor with a ferromagnet. They show that the Tellegen effect in this metamaterial is larger by two orders at visible frequencies and four orders at far-infrared than in natural materials. This large effect is very impressive and potentially useful for applications in spintronics. The possibility of locally tuning the Tellegen coefficient is. Moreover, the authors' construction is creative and simpler than a naïve construction based on Tellegen meta-atoms, which itself has zero magnetization and zero electric dipole moment. The authors used the fact that the magnetization and electric dipole moment of a single meta-atom does not need to be zero in a randomly oriented array, because those moments cancel out on average, while the Tellegen coefficient averages to a finite value. The manuscript is clearly written and easy to read. I recommend this paper for publication in Nature Communications.

Reply: We thank the reviewer for the thoughtful and positive evaluation of our work.

1) It is unclear why the meta-atom has a parity-time (PT) symmetry. Does the parity mean mirror parity or spatial-inversion parity?

Reply: We thank the reviewer for pointing out this terminology issue. The statement in the sentence that read "Thus, the meta-atom has a parity-time (PT) symmetry required for Tellegen metamaterial constituent" was misleading. What we meant is that the meta-atom indeed breaks parity symmetry and reciprocity, as required for Tellegen metamaterial constituent. We have corrected this sentence to: "Thus, both parity (space-inversion) symmetry and reciprocity are broken in the meta-atom, as required for a Tellegen metamaterial constituent." (see page 5). Please note that in this context, parity refers to spatial-inversion operation.

Referee 2

The authors describe an interesting proposal to realize an isotropic Tellegen response based on a particle with two components: a piece with permanent magnetization (responsible for the nonreciprocal response) and a resonant Si cylinder supporting a magnetic Mie resonance. They show that by symmetry such a nanoparticle can have a Tellegen uniaxial response (as well as other types of dipolar couplings, such as a gyrotropic-type response). They argue that after the random mixing of the nanoparticles, only the Tellegen and chiral isotropic pieces can survive due to symmetry (the chiral term vanishes by symmetry). They propose two designs that yield a sizable Tellegen response at optical frequencies. Furthermore, they nicely highlight the role of the parity and time-reversal symmetries, showing that only designs that break simultaneously the two symmetries, but that respect the PT symmetry can give the desired material response. On the overall, this is nicely written and timely article and I support its publication in Nat. Communications, essentially in its present form. The article is important as, to my knowledge, it is the first fully-fledged proposal of a Tellegen-type metamaterial at optical frequencies. Tellegen responses are interesting conceptually and for nanophotonic applications.

Reply: We thank the reviewer for such a positive response to our work.

1) Page 3: it is strange to designate α_{χ_a} as the anti-symmetric Tellegen piece as by definition a Tellegen response must be a symmetric tensor. Does α_{χ_a} describe a moving-medium coupling?

Reply: We thank the reviewer for pointing this out. As the reviewer correctly mentions, α_{χ_a} corresponds to the so-called “synthetic moving” response of the meta-atom, whereas α_{κ_a} describes the so-called omega coupling, respectively. To enhance clarity, the previous sentence has been revised in page 3: “Polarizabilities $\alpha_{\chi_s} = \alpha_{\chi}^{xx} = \alpha_{\chi}^{yy}$ and α_{χ_a} denote the Tellegen and so-called “synthetic moving” response of the meta-atom, whereas $\alpha_{\kappa_s} = \alpha_{\kappa}^{xx} = \alpha_{\kappa}^{yy}$ and α_{κ_a} describe its chirality and the so-called omega coupling, respectively.”

2) Page 5, Fig.2(a): the figure shows a plot of H_x . Is this really the relevant quantity to show? (the induced magnetic dipole is along-y). Perhaps it would be better to show the electric currents in the considered plane?

Reply: We apologize for the typos in Figs. 2(a) and 3(a), where H_x was labeled. We have corrected it to H_y , which is actually plotted in this figure.

3) Page 5: in the design of Fig. 2 what controls the material loss? The intrinsic loss of Co? Or the loss of Si? Radiation loss due to the random mixing? Please comment on the article.

Reply: Indeed, both the absorption loss in cobalt and silicon nanodisks and the radiation loss due to the random mixing contribute to the imaginary part of permittivity and permeability in Figs. 2(c,d). We have commented on this in the main text (page 6): “The nonzero imaginary parts of ϵ_{eff} and μ_{eff} are due to the absorption loss in cobalt and silicon nanodisks as well as the radiation loss due to the random mixing of the meta-atoms in the metamaterial.”

4) Page 8: the article ends by arguing that the considered developments may lead to experimental implementations of concepts related “axion matter, dyons, Witten effect, etc”. I recommend that if there is indeed some interesting optics analogy/implication that it is elaborated a bit more for one of the particular items. I do not find specially useful/appealing to present a long list of items that appear to be completely detached from any specific physical setup/electromagnetic effect that can be understood by a reader.

Reply: We thank the reviewer for this comment. We have added a discussion on the optical implication of the dyon quasiparticles in Tellegen metamaterials (see page 9). Specifically, it reads: “For example, by positioning an electric monopole charge near a bulk Tellegen metamaterial, one can excite a peculiar charge distribution inside the metamaterial that would correspond to a combination of effective electric and magnetic monopole charges (dyon charge) [81].”

Referee 3

The manuscript authored by S.S. Jazi et al. describes a possible physical implementation of an isotropic Tellegen medium operating at optical frequencies. This is achieved through a meta-atom design comprising a spontaneously magnetized ferromagnetic nanocylinder attached to a dielectric nanocylinder. The primary idea behind this research is to use the gyroelectric response displayed by the ferromagnetic nanocylinder to excite a Mie-resonant magnetic mode within the dielectric nanocylinder. Although the gyrotropic electric dipole response is perpendicular to the high-frequency electric field, the magnetic response of the dielectric nanocylinder induced by the gyroelectric dipole aligns with the electric field. Because of that and due to the non-

reciprocity of the gyroelectric response, the meta-atom exhibits characteristics similar to those of an anisotropic Tellegen particle. A metamaterial composed of such randomly oriented meta-atoms forms an isotropic Tellegen medium.

In the section discussing existing proposals for realizing Tellegen media targeting microwave frequencies, the authors assert that the main challenge in such realizations lies in creating a local magnetic biasing field. However, I disagree with this viewpoint, as the mentioned difficulty could be effectively addressed by equipping each meta-atom with a small permanent magnet attached to it. Consequently, I suggest that the manuscript authors could improve this discussion by outlining other - and possibly more important - differences between their proposed implementation and the existing approaches

Reply: We thank the reviewer for the overall very positive evaluation of our work. We agree with the comment about the possibility of using permanent magnets for the microwave realizations of Tellegen metamaterials. Indeed, although it was never proposed in the literature to the best of our knowledge, we imagine that it could serve as a viable solution in the microwave frequency range. We have corrected our statement accordingly (see page 4): “Nevertheless, local magnetization of microwave meta-atoms is, in principle, possible if they are equipped with non-conducting permanent magnets, such as barium ferrite $\text{BaFe}_{12}\text{O}_{19}$, but such implementations have not yet been reported.” We have also extended the discussion about the difference between the proposed and the existing optical implementation (see the new paragraph in page 4):

“Our method of achieving local magnetization through the anisotropies of nanocylinders is drastically different from the approach taken in the recently proposed uniaxial multilayer Tellegen metamaterial [19]. Achieving local magnetization in such a multilayer film, which features an antiferromagnetic out-of-plane configuration, presents a considerably greater challenge because maintaining opposite magnetization vectors in neighboring domains is energetically unfavorable.”

Additional changes

- The vertical axis labels in Figure 2e contained errors, which have now been corrected. Importantly, these errors did not affect the accuracy of the results presented in the text.
- We have added two new relevant references [25,43].

Sincerely yours,

The authors

REVIEWERS' COMMENTS

Reviewer #1 (Remarks to the Author):

The authors addressed my question satisfactorily. Their responses to other reviewer comments also look convincing. I recommend this paper for publication in Nature Communications in its current form.

Reviewer #2 (Remarks to the Author):

I have read the replies to the referees and the revised article. Regarding my previous question on "what controls the material loss" the question was what is the *dominant* loss channel? The intrinsic loss of Co? Or the loss of Si? Radiation loss due to the random mixing? Please comment on the article.

Reviewer #3 (Remarks to the Author):

In the revised version of the manuscript, the authors addressed my comments and implemented necessary corrections. I recommend this manuscript for publication.